# DATA-DRIVEN EVALUATION OF TRAINING ACTION SPACE FOR REINFORCEMENT LEARNING

## ABSTRACT

Training action space selection for reinforcement learning (RL) is conflict-prone due to complex state-action relationships. To address this challenge, this paper proposes a Shapley-inspired methodology for training action space categorization and ranking. To reduce exponential-time shapley computations, the methodology includes a Monte Carlo simulation to avoid unnecessary explorations. The effectiveness of the methodology is illustrated using a cloud infrastructure resource tuning case study. It reduces the search space by 80% and categorizes the training action sets into dispensable and indispensable groups. Additionally, it ranks different training actions to facilitate high-performance yet cost-efficient RL model design. The proposed data-driven methodology is extensible to different domains, use cases, and reinforcement learning algorithms.

## 1 INTRODUCTION

A reinforcement learning (RL) agent learns how to map states to actions in order to maximize a long-term cumulative award signal in a given environment. Figure 1 shows the various artifacts of an RL algorithm. An RL problem is defined by a quartet of $(S, A, P_a, R_a)$, where $S$ is a set of states or the state space; $A$ is a set of actions or the action space available to influence $S$; $P_a(s, s') = Pr(s_{t+1} = s' | s_t = s, a_t = a)$ is the transition probability which is the probability that action $a$ in state $s$ at time $t$ will lead to state $s'$ at time $t + 1$; and finally, $R_a(s, s')$ is the immediate reward signal received after transitioning from state $s$ to state $s'$, due to action $a$. An RL agent training involves recognizing an optimal policy function, $\pi^* : a \leftarrow s$, from a corpus of $\{(s_i, a_i)\}$ to maximize the long-term cumulative reward, $\sum R_a$. The reward function, $R_a$, is defined ab initio for an efficient goal accomplishment. The transition probability or state-action mapping is defined by the environment dynamics. In many real-life use cases, the agent cannot directly sense the effect of its actions on the environment. This is particularly true when the state-action relationship cannot be modeled by either closed-form analytical expressions [26, 29, 58] or explicit rules as in the popular games such as Chess [28], Go [47, 46], and Atari [34]. This challenge is well documented in the literature [51, 57]. To address this challenge for RL model training [52, 33], simulation-based action models [50] play a pivotal role. The optimal choice of simulation parameters such as the training action space is a non-trivial challenge because of the curse of dimensionality [17], non-linearity [11], and non-uniform action set [25].

Training data valuation [22, 10] and associated artisanal software engineering efforts constitute a large part of the machine learning (ML) life cycle (or MLOps). Yet, most research and development efforts [36] focus on algorithms and infrastructure. Production-grade MLOps needs to handle data lifecycle management (DLM) [40] challenges including: fairness and bias in labeled datasets [16], data quality [9], limitations of benchmarks [42], and reproducibility concerns [39]. For RL, the DLM challenges are further compounded due to complexities arising from non-linear state-action interactions, partially-observable processes, non-isometric action spaces [32], and strong domain specificity [21] of action models. With the recent emphasis on ML explainability, traditional supervised learning and deep learning research communities are actively working on systematic data-driven frameworks [14, 53] for training data valuation. We need equivalent frameworks for RL to streamline conflict-prone training action selection [8, 37, 18]. Depending on the agent-environment interactions, different training actions have different relative contributions to the RL agent performance. Some training actions are indispensable because of their unique positions in the parametric space. Other dispensable actions have different relative contributions to the reward function. *Remarkably, a high-cardinality*

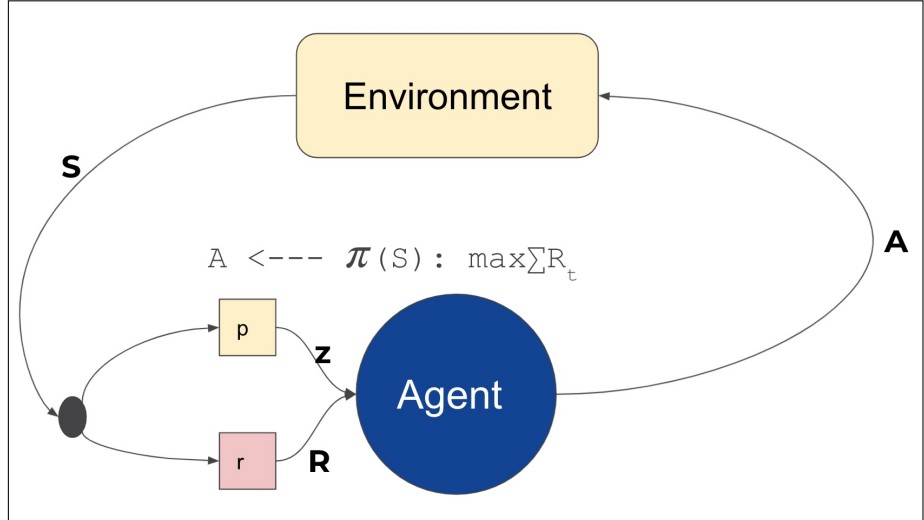

Figure 1: The schematic diagram for a typical reinforcement learning algorithm

*training action space in many cases leads to lower cumulative reward than a well-designed training action space.* To the best of authors' knowledge, there is hardly any data-driven tool for the evaluation of training action space for RL. Such a tool leads to superior RL agent performance, lower model training and maintenance cost, and strong multi-disciplinary collaboration [35].

This paper proposes a shapley-inspired [54] algorithm to categorize and rank training action sets. It also assists in recognizing cut-off cardinality for the training action space to reduce unnecessary exploration and ensure polynomial time complexity. While Section 2 describes the algorithms for RL training action space evaluation, Section 3 illustrates the effectiveness of the algorithms in a specific case study. Section 4 discusses the relevant work. Finally, Section 5 presents a summary with possible future directions.

## 2 ALGORITHM

The proposed framework provides a shapley-inspired methodology [49] to efficiently filter out unnecessary training actions and discover a near-optimal action set. Figure 2 shows the framework consists of two algorithms and how these two algorithms interact with each other to design a near-optimal RL model by training action space filtering and evaluation. The first algorithm computes cut-off cardinality for the training action space. The cut-off cardinality is defined as the minimium number of action points needed in a training action set to enable high-fidelity predictions within an acceptable computational complexity and an error bound. The second algorithm takes action points only above the cut-off cardinality and categorize action points into two classes: $\{dispensable, indispensable\}$ and ranking dispensable training action points based on the corresponding cumulative rewards. Indispensable action points are absolute essential action points for the RL training to achieve the desired goal. On the other hand, dispensable action points are non-essential action points that can be discarded. However, the RL agent might perform sub-optimally without a disposable action point. By ranking disposable training action points based on the corresponding cumulative rewards, the relative contributions of different training action points are estimated.

### 2.1 CUT-OFF CARDINALITY COMPUTATION FOR TRAINING ACTION POINTS

RL deals with compilation of a quartet $(S, A, P_a, R_a)$ and computation of an optimal policy, $\pi^*$ which maximizes the long-term cumulative reward. The search space for a shapley-inspired technique needs to span across the power set for the training action space which has exponential complexity with training action points. To circumvent this problem, the proposed algorithm, as discussed in Algorithm 1, uses Monte Carlo method to compute the cut-off cardinality. The inputs to this algorithm include an RL algorithm, the power set for the training action space, the corresponding state space,

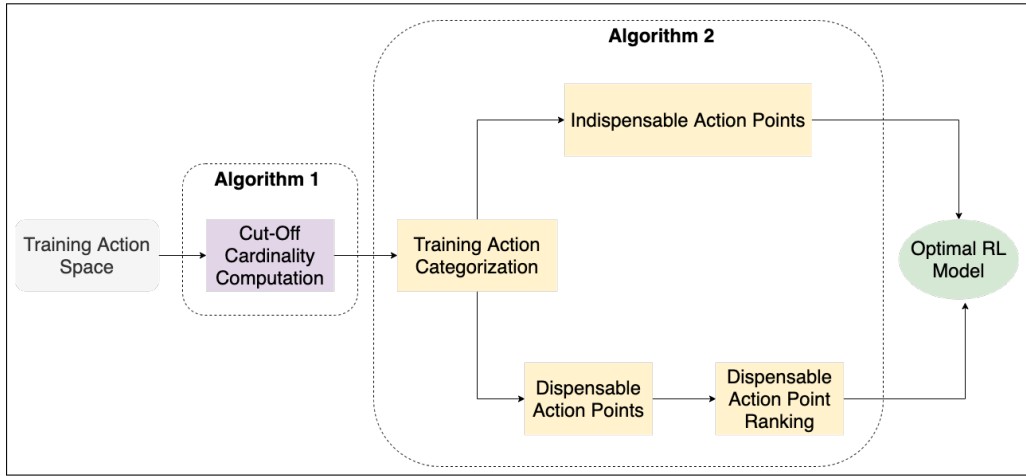

Figure 2: The proposed training action selection framework for reinforcement learning

the transition probability, an acceptable performance margin, and an acceptable number of iterations. The output is the cut-off cardinality. The algorithm uses an iterative procedure to train the RL agent with different numbers of training actions, starting from all available training actions to only one training action. If for a certain cardinality, all the cumulative rewards do not improve by $\epsilon$ within $n$ iterations, then we set the status to failure, and assign the cut-off cardinality to that cardinality plus one. The cut-off cardinality signifies computational efficiency — a larger cut-off cardinality means lower computational overhead and vice versa.

---

**Algorithm 1** Monte Carlo method for cut-off cardinality computation for the training action space

---

**Input:** The RL algorithm, $\mathcal{R}$, the power set for the training action space, $\mathcal{P}(A)$, the corresponding state space, $\mathcal{P}(S|A)$, transition probability $\boldsymbol{P}$, an acceptable performance margin, $\epsilon$, and an acceptable number of iterations, $n$.
**Output:** Cut-off cardinality, $|A|_{cutoff}$
    **for** $k \leftarrow |A|$ to 1 **do**
        **repeat**
            $\pi^* \leftarrow \mathcal{R}(S, A, P_a, R_a)$
            **if** the cumulative reward improves by $\epsilon$ with $n$ iterations **then**
                $status \leftarrow success$
            **else**
                $status \leftarrow failure$
            **end if**
        **until** all necessary training action sets for a given cardinality is exhausted.
        **if** all statuses for a given cardinality, $k$, are $failures$ **then**
            $|A|_{cutoff} \leftarrow k + 1$
        **end if**
    **end for**

---

## 2.2 CATEGORIZATION AND RANKING OF TRAINING ACTIONS

Algorithm 2 offers a principled procedure for training action selection. As for inputs, it takes the combinations of training state-action pairs above the cut-off cardinality, determined by Algorithm 1, the RL algorithm, the reward function, the threshold condition, and the acceptable number of iterations. As for outputs, it predicts a categorization vector which classifies training action points into two classes: {*dispensable*, *indispensable*} and ranks dispensable training actions based on the corresponding cumulative reward. At its core, Algorithm 2 determines whether the policy determined from the given state-action pairs achieves the threshold condition within the acceptable number of iterations. The algorithm uncovers the indispensability of some training action sets for a given RL task. A training action set is deemed to be indispensable: *iff* upon removal of that training action

point, the RL agent fails to accomplish the goal within a finite number of iterations [24, 56, 23]. All other action points are dispensable. The dispensable action points are rank listed in $\mathcal{L}$ based on the corresponding cumulative reward. Essentially, $\mathcal{L}$ informs the performance impact of different disposable action points on the RL agent.

---

**Algorithm 2** Data-driven categorization and ranking for training action space evaluation

---

**Input:** All combinations of training state-action pairs upto cut-off cardinality: $\{(s_i, a_i)\}$, the RL algorithm, $\mathcal{R}$, the reward function, the threshold condition, and the acceptable number of iterations.
**Output:** Categorization vector, $\mathcal{V}$, which classifies training action points into two classes: {*dispensable*, *indispensable*}. Rank order list of dispensable training actions based on the corresponding cumulative reward.
   **repeat**
      **for** $< s_i, a_i >$ **do**
         $\pi^* \leftarrow \mathcal{R}(S, A, P_a, R_a)$
         **if** Threshold condition is *never* satisfied *within* the acceptable number of iterations **then**
            $\mathcal{V}(\boldsymbol{A} - a_i) \leftarrow indispensable$
         **else**
            $\mathcal{V}(\boldsymbol{A} - a_i) \leftarrow dispensable$
         **end if**
      **end for**
   **until** all necessary training action sets are evaluated.
   Rank List, $\mathcal{L} \leftarrow$ dispensable actions sorted in order of the corresponding cumulative rewards

---

## 2.3 ACTION SHAPLEY

The proposed framework adopts Action Shapley which introduces a formulation for the problem of equitable training action valuation in reinforcement learning. Reinforcement learning has three key building blocks. The first building block is the training action space, $A$, the corresponding state space $S$, and a pre-assigned reward function, $R_a$. The second building block is a reinforcement learning algorithm, $\mathcal{R}$, which is treated as a black box. It takes $(S, A, P_a, R_a)$ and computes an optimal policy function, $\pi^*$. The third building block is the performance measure or the cumulative sum of reward, $\sum R_a$. The goal of the framework is to compute the valuation for each training action point, $\phi_i(A, \mathcal{R}, \sum R_a)$ as a function of three building blocks. A shapley technique is useful for action valuation because it satisfies nullity, symmetry, and linearity [45]. From game theory, $\phi$ can be expressed as shown in Equation 1:

$$\phi_i = C \sum_{S \subseteq A - \{i\}} \frac{\sum R_a(S \cup \{i\}) - \sum R_a(S)}{\binom{n-1}{|S|}} \tag{1}$$

where, the sum is over all subsets of $A$ not containing $i$ and $C$ is an arbitrary constant. We call $\phi_i$ the Action Shapley value of action $a_i$. Equation 1 suggests Action Shapley computation is with exponential time complexity w.r.t. the training action points. This exponential computational complexity warrants the need for the cut-off cardinality discussed in Algorithm 1.

## 3 CASE STUDY

In this section, we design a resource tuning example (in the cloud) to illustrate the effectiveness of the proposed algorithm for the training action space evaluation. We evaluate the performance of an MDP [20]-based RL agent, as shown in Figure 1. The state-action mapping and transition probability are modeled using time-series auto-regression [38] after a PCA-based dimensionality reduction [41] with time complexity $\mathcal{O}(k log k)$, where $k$ is the number of principal components. The choice of the algorithms is purely driven by the nature of the training dataset. A more complex non-linear dataset warrants more complex sequential models such as long short-term memory (LSTM) [44] or attention based models [30]. For action updates, WLOG, an RL-based PID controller [1, 13] is used. Similar to time-series modeling, action update can be conducted by other policy learning algorithms such as SARSA [19].

Table 1: Relevant case study for cloud resource tuning

| RL Artifact | Description |
|---|---|
| *State* | CPU utilization metrics |
| *State Statistics* | Median value of CPU utilization |
| *Threshold* | 90% of CPU utilization |
| *Action* | Resource configuration set points: (# of vCPUs, Memory Size (GB)) |
| *Reward* | Negative of total time steps required to satisfy the threshold condition |
| *Parametric Boundary* | Polygon defined by the parametric endpoints |
| *Initial Action* | $(6, 14)$: Arbitrarily assigned WLOG |
| *Error Margin* | 5% |
| *Acceptable Steps* | 400 |
| *Computational Complexity* | Polynomial time |

Table 2: Five different AWS EC2 resource pairs used in training action space

| EC2 Type | of vCPUs | Memory Size (GB) |
|---|---|---|
| *small t3a* | 2 | 2 |
| *medium t3a* | 2 | 4 |
| *large t3a* | 2 | 8 |
| *xlarge t3a* | 4 | 16 |
| *2xlarge t3a* | 8 | 32 |

As shown in Table 1, the objective of the RL agent for this case study is to quickly reduce high CPU utilization below a pre-assigned threshold for a given workload. In most infrastructure/cloud resource tuning technologies [55], CPU utilization represents a key metric. Therefore, the state space for this RL case study consists of virtual machine CPU utilization (%) metrics and the action space is defined by the VM resource set points, i.e., (# of vCPUs, Memory Size (GB)). The RL agent uses AWS boto3 SDK [2] to manipulate actions and AWS CloudWatch [3] for state space monitoring. The reward is defined as the number of time steps required by the agent to accomplish the objective multiplied by $-1$. The negative reward per time step was meant to push the agent to accomplish the task as fast as possible.

The training data for this case study was generated internally [4] with an open source library, *stress* (https://linux.die.net/man/1/stress). It uses a rectangular workload. The peak of the workload uses the stress command: *sudo stress –io 4 –vm 2 –vm-bytes 1024M –timeout 500s*. Essentially, the peak is running 4 *I/O* stressors and 2 VM workers spinning on malloc with 1024 MB per worker for 500 s. The simulated rectangular workload has a time period of 600 s: a high-stress phase of 500 s is followed by an inactive phase of 100 s. The RL training action space is spanned by the power set drawn from the five pairs of EC2 configuration set points as shown in Table 2. Using Algorithm 1, we notice that the power set below the cut-off cardinality of 4 produces trivial and unstable results. Therefore, WLOG, the analysis in this paper has been focused on six training action sets as shown in Table 4. That amounts to 81.25% reduction in the search space. For each EC2 instance in the training action space, the corresponding state metrics, i.e., CPU utilization (%) are shown in Figure 3. The training data was collected for a 24 hour period with 1 minute sampling interval. Using Algorithm **??**, the optimal step-size for action update is identified to be 0.1 in both # of vCPUs and Memory Size (GB) dimensions.

With this set up, different RL models are developed with different training action sets and the corresponding rewards and ranks are noted in Table 4. Remarkably, a high cardinality training action set does not guarantee the best reward: the agent with all training actions,<*small t3a, medium t3a, large t3a, xlarge t3a, 2xlarge t3a*>, does not yield the highest reward. In fact, the action set with <*medium t3a, large t3a, xlarge t3a, 2xlarge t3a*> yields the highest reward. This pattern could be attributed to the state-action interaction in this particular case study. First, the parametric distances

Table 3: Training action categorization based on the valuation vector

| Training Action | Category |
|---|---|
| *small t3a* | dispensable |
| *medium t3a* | dispensable |
| *large t3a* | indispensable |
| *xlarge t3a* | indispensable |
| *2xlarge t3a* | indispensable |

Table 4: Rewards and ranks for different training actions. Some training action sets fail to satisfy the objective. Therefore, the rewards and ranks for them are noted as *none*
.

| Training Action Set | Reward | Rank |
|---|---|---|
| *<small t3a, medium t3a, large t3a, xlarge t3a, 2xlarge t3a>* | $-21$ | 3 |
| *<medium t3a, large t3a, xlarge t3a, 2xlarge t3a>* | $-13$ | 1 |
| *<small t3a, large t3a, xlarge t3a, 2xlarge t3a>* | $-16$ | 2 |
| *<small t3a, medium t3a, xlarge t3a, 2xlarge t3a>* | *none* | *none* |
| *<small t3a, medium t3a, large t3a, 2xlarge t3a>* | *none* | *none* |
| *<small t3a, medium t3a, large t3a, xlarge t3a>* | *none* | *none* |

between different EC2 instance pairs are not uniform. While the Euclidean distance between *small t3a* and *medium t3a* is equal to 2, that between *xlarge t3a* and *2xlarge t3a* is 16.5. The non-uniform spacing for training action space is a considerable deterrent [31] for RL adoption. Second, in this case study, the transient CPU utilization (%) patterns have undergone a material change from *xlarge t3a* (max 100%) to *2xlarge t3a* (max 73%). This indicates the strong influence of *2xlarge t3a* for the given RL task. Indeed, we noticed *2xlarge t3a* to be an indispensable training action. As shown in Table 3, a categorization of training actions can be inferred based on the valuation vector, $\mathcal{V}$, as described in Algorithm 2: two *dispensable* training actions are uncovered to be *small t3a, medium t3a* and three *indispensable* training actions to be *large t3a, xlarge t3a, 2xlarge t3a*.

Figure 4 shows the RL loop action with three training actions: *<small t3a, medium t3a, large t3a, xlarge t3a, 2xlarge t3a>*, *<medium t3a, large t3a, xlarge t3a, 2xlarge t3a>*, and *<small t3a, large t3a, xlarge t3a, 2xlarge t3a>*.

- For the first training action set of *<small t3a, medium t3a, large t3a, xlarge t3a, 2xlarge t3a>*, the reward is $-21$. The left subplot in Figure 4(a) shows how the recommended action is evolving with time from an *arbitrary* initial point of (6, 14). The recommended points are superimposed on the training action parameter space to illustrate their relative position with respect to the parametric boundary which is defined by the trapezium with vertices: *{(2,2), (2,8), (8,32), (8,2)}* in the (# of vCPUs, Memory Size (GB)) space. The right subplot in Figure 4(a) shows how the median CPU utilization (%) comes below the *90%* threshold in 21 steps leading to $-21$ in reward. The blue dots represent the median CPU utilizations for different training set points.

- For the second training action set of *<medium t3a, large t3a, xlarge t3a, 2xlarge t3a>*, the reward is $-13$ as shown in Figure 4(b).

- For the third training action set of *<small t3a, large t3a, xlarge t3a, 2xlarge t3a>*, the reward is $-16$ as shown in Figure 4(c).

As shown in Table 4, the reward scores can indeed be used for ranking different training action sets leading to a data-driven approach for training action selection. As shown in Figure 5, *large t3a* is an indispensable element in the training action space for the given RL agent. Without this training action, the RL agent fails to accomplish the goal of bringing the CPU utilization below the critical threshold. Similar observations can be made about *xlarge t3a* and *2xlarge t3a*.

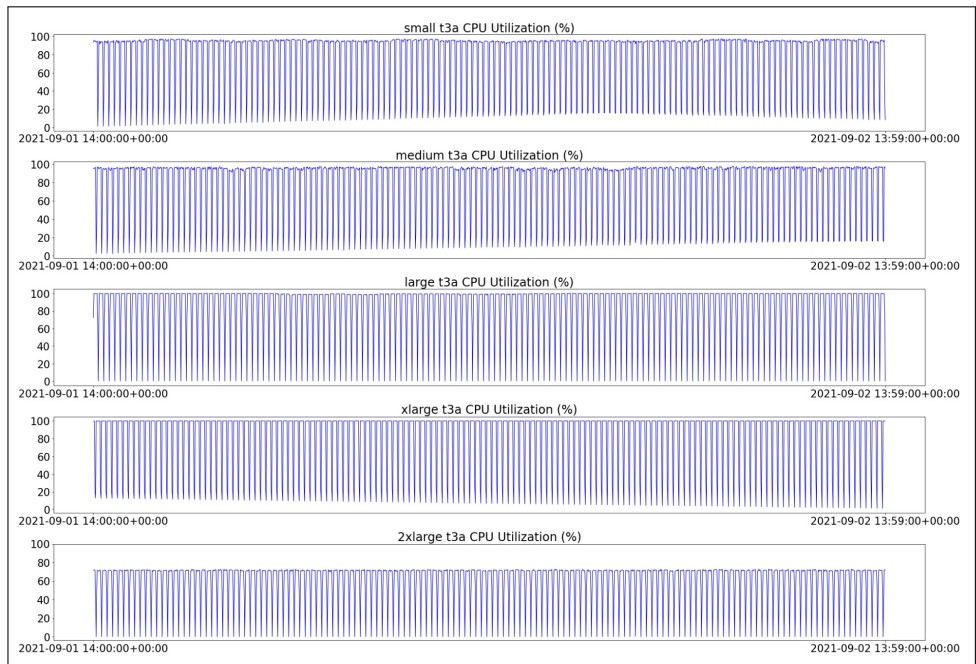

Figure 3: CPU utilization (%) responses on five different EC2 instances (Table 2) from 2PM-UTC 9/1/2021-2PM-UTC 9/2/2021 at 1 minute granularity. This training data was generated internally [4]. The median CPU utilization values (%) are noted to be equal to {95%, 95.5%, 99.5%, 100%, 72.5%}
.

## 4 RELATED WORK

Shapley value was postulated in a classic paper in game theory [15] and influences the field of economics significantly [43]. Various shapley-inspired techniques have been applied to model diverse problems including voting [12], resource allocation [48] and bargaining [27]. Recent years saw increased application of shapley-inspired methodologies in machine learning feature importance evaluation and training data valuation [14, 10]. To the best of our knowledge, Shapley value has not been used to quantify training action valuation in a reinforcement learning context. The application of shapley-inspired technique for reinforcement learning is a challenging problem. First, the relationship between state-action pairs and transition probabilities are expensive to model. Also, often the RL training action points are not uniformly positioned. Finally, we have purposefully focused on a small sample cloud infrastructure use case instead of cliched RL use cases [5]. That is because the problem of optimal action selection has relatively more expensive in the cloud infrastructure space where the state-action relationship is only partially observable.

## 5 CONCLUSION

This paper proposes a data-driven methodology for training action space evaluation for RL. The methodology offers a principled framework for training action space categorization and ranking within a finite computational time. It unleashes a strategy for superior model performance and lower modeling cost. Additionally, the proposed methodology is completely agnostic of use cases and machine learning algorithms. Therefore, it is a general-purpose methodology extensible to different domains including distributed computing, network traffic control, healthcare, automatic locomotion, building management system, and industrial controls, and different machine learning algorithms such as PCA, Autoencoder, ARIMA, LSTM, Transformer, PID, SARSA, DQN, and many others. For the next phase of the development for this data-agnostic methodology, we are planning to contribute a general RL design library to relevant open source projects [6, 5, 7].

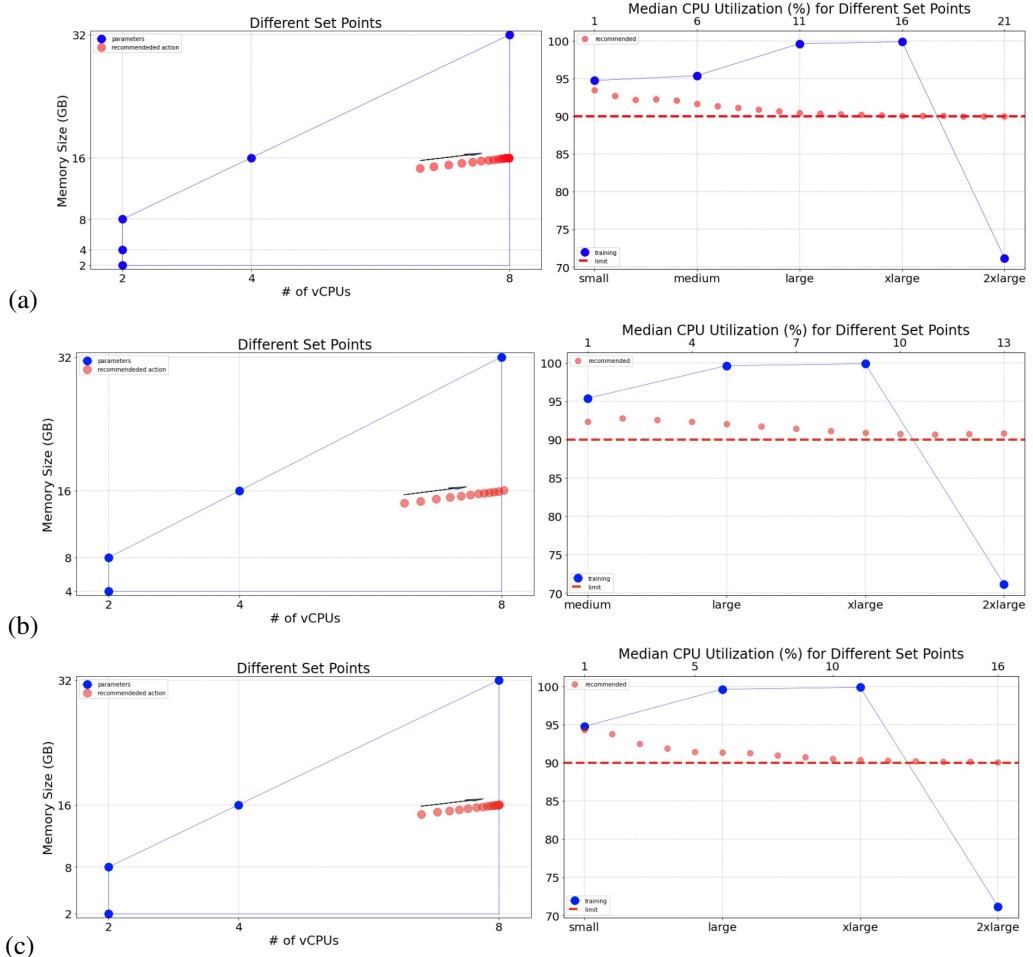

Figure 4: Examples of dispensable actions. (a) RL loop with all training actions, <*small t3a, medium t3a, large t3a, xlarge t3a, 2xlarge t3a*>. The reward is noted to be $-21$. (b) RL loop with a training action set of <*medium t3a, large t3a, xlarge t3a, 2xlarge t3a*>. The reward is noted to be $-13$. (c) RL loop with a training action set of <*small t3a, large t3a, xlarge t3a, 2xlarge t3a*>. The reward is noted to be $-16$. Both *small t3a* and *medium t3a* are noted to be dispensable actions.

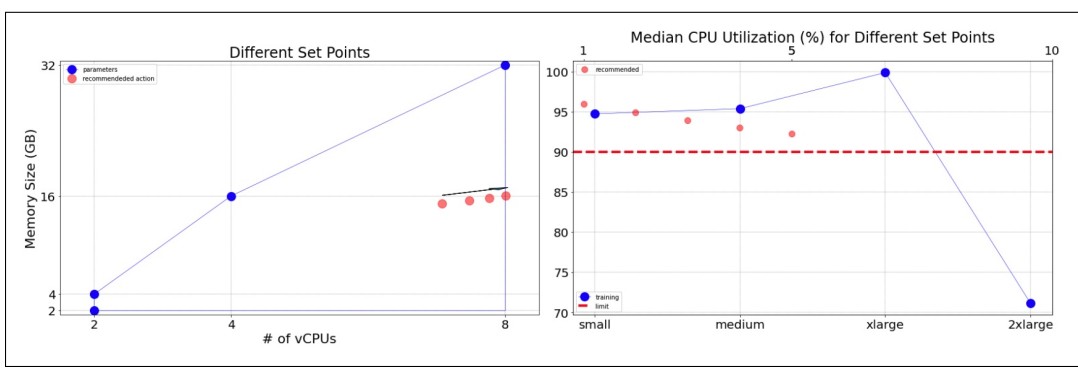

Figure 5: RL loop with <*small t3a, medium t3a, xlarge t3a, 2xlarge t3a*> action space. The agent could never accomplish the goal, therefore, *large t3a* is an indispensable element in the training action space. Similarly, *xlarge t3a* and *2xlarge t3a* are two indispensable elements in the training action space.

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

## 6 NOTATION

| | |
|---|---|
| $S$ | Space Space |
| $A$ | Action Space |
| $P_a$ | Transition Probability |
| $R_a$ | Reward Signal |
| $\mathcal{P}$ | Power Set |
| $\pi$ | Policy |
| $\pi^*$ | Optimal Policy |
| $\pi^*$ | Optimal Policy |
| $\mathcal{V}$ | Categorization Vector |
| $\mathcal{L}$ | Rank List of Dispensable Actions |

