# OpenReview forum: "DATA-DRIVEN EVALUATION  OF TRAINING ACTION SPACE FOR REINFORCEMENT LEARNING"
_ICLR.cc/2022/Conference — ICLR 2022 Submitted_

### Official Review · Reviewer_5emL · 2021-10-25

**Correctness:** 2
**Technical Novelty And Significance:** 1
**Empirical Novelty And Significance:** 2
**Recommendation:** 3
**Confidence:** 4

**Main Review:**

While understanding the impact of training action space is certainly a valid and important problem, the investigation of this topic is far from satisfactory in the paper.

First, the design of the new algorithm is mainly presented in the pseudo-code form without further explanation. It is not easy to follow the new algorithm design and understand why the new algorithm can address the optimal training action selection problem. The authors mentioned that their algorithm is shapely-inspired. However, there is no further discussion regarding how this idea is actually utilized in the algorithm design and why the new algorithm is novel.

Second, many key concepts explored in the paper are not well defined. For example, it is quite confusing to me what the dispensable action set and indispensable action set truly mean. The corresponding discussion in the paper is brief. The authors also stated that "the possible action space grows exponentially with the number of training actions". I can hardly understand why the action space grows exponentially with the number of actions. This does not make any sense.

Third, I am not sure why the authors choose to evaluate their methods on a specific case study related to cloud computing. Why is this case study particularly important for understanding the impact of training action space on reinforcement learning performance? Why didn't the authors consider some of the most commonly used benchmark reinforcement learning problems?

Finally, the experiment results are quite limited and flawed. No solid conclusions can be derived from the experiment results. Hence the overall technical contributions of this paper are highly questionable. The clarity and quality of this paper should also be substantially improved.

**Summary Of The Paper:**

This paper empirically considered the impact of training action space for reinforcement learning in a case study. Understanding the impact of training action space is a valid and important problem. An empirical study of this problem appears to be the main contribution of this paper.

**Summary Of The Review:**

This paper has major weakness in several aspects, including both the algorithm design and experimental analysis. The paper is not ready for publication with its current shape.

---

> ### Author Response · Authors · 2021-11-23
> **Implemented your suggestions**
>
> Your feedback is greatly appreciated. To address your points, the following changes have been made:
>
> 1. Algorithm section has been thoroughly revised.
>
> 2. We thought cloud infrastructure use case is most relevant because this is a practical use case where action space is expensive to compile and state-action relationships are hard to model.
>
> 3. While the experimental results appears to be limited, this represents the practical MLOps problem where training data is hard to compile and the system is partially observable. We wanted to solve a small sample RL problem.

---

> > ### Comment · Reviewer_5emL · 2021-11-28
> > **Thank you for your response**
> >
> > Thanks for revising the paper and providing response to my comments. However, I am afraid that some major concerns from other reviewers and me remain unanswered. Hence I will keep my initial rating.

---

### Official Review · Reviewer_ASBS · 2021-11-01

**Correctness:** 2
**Technical Novelty And Significance:** 2
**Empirical Novelty And Significance:** 2
**Recommendation:** 1
**Confidence:** 4

**Main Review:**

While the paper is well motivated, the current state of this work is unsatisfactory. In fact, this paper appears to be severely incomplete (the manuscript is 7 pages long, with no dedicated related work section).

Firstly, the method consists of 3 algorithms which are not thoroughly explained. Indeed, while the authors introduce the method with a good overview in Section 2, the technical discussion of the framework is lacking. For example, the authors say their method is “Shapely inspired”, yet they do not say why and in what sense. Although the algorithms appear simple and are provided in full with pseudo-code, they should be explained in detail in a Machine Learning paper.

Secondly, the evaluation of the method is rather limited. The framework is evaluated on a single case study, where the dataset was generated internally on purpose for this work, i.e. the method was not evaluated on a standard benchmark already available to the community (besides, a GitHub link to such data is provided, which potentially breaks anonymity). The action space of such case study is also fairly limited, where the number of actions is 2: number of CPUS and RAM size in GB. While results show the method is effective, such evaluation hardly challenges the premise of the paper where complex and large action spaces are concerned.

Important results analysis is missing too. For example, what is the computational cost of the method? Since the aim of this work is to reduce action space search in order to achieve a better policy and save computing resources, the additional overhead of the framework should be assessed. This point is missing altogether.

Finally, no other baselines are provided at all, thus it is impossible to properly judge the performance of the method.

**Summary Of The Paper:**

This paper focuses on Reinforcement Learning (RL). RL methods often entail a potentially large action space exploration to find a good policy. This work proposes a method to reduce such action space exploration. The method separates actions into two categories: dispensable (the action can be ignored) and indispensable (the action must be taken). Dispensable actions are also ranked according to their importance with respect to the final policy. The method is data driven and operates by looking at the global reward returns obtained when a certain action is removed from the action space. The method is evaluated on a case study simulating cloud infrastructure workload optimisation, i.e. the task of reducing high CPU utilisation by allocating more resources.

**Summary Of The Review:**

This is an incomplete work. The method is not explained in detail and is not evaluated in a sufficient manner.

---

> ### Author Response · Authors · 2021-11-23
> **Added the relevant sections**
>
> Your feedback is greatly appreciated. To address your points, the following changes have been made:
>
> 1. A related work section has been added.
>
> 2. The algorithm section has been thoroughly revised.
>
> 3. To our knowledge, there is no benchmark in the RL community.
>
> 4. Indeed, the action space is somewhat sparse. But, this is a practical problem in the cloud infrastructure.
>
> 5. We thought it is quite obvious that the cut-off cardinality means we are reducing the search space and saving on the computational overhead. If you advise, we can add a detailed discussion in the appendix.

---

> > ### Comment · Reviewer_ASBS · 2021-11-25
> > **Thanks, but still below acceptance threshold**
> >
> > Thanks for providing an updated version of the manuscript. I'm afraid though that most of the concerns the other reviewers and I have raised remain unanswered. I believe the overall quality of this work is not good enough for ICLR, thus I keep my initial rating.

---

### Official Review · Reviewer_TWbe · 2021-11-02

**Correctness:** 1
**Technical Novelty And Significance:** 1
**Empirical Novelty And Significance:** 1
**Recommendation:** 1
**Confidence:** 4

**Main Review:**

My understanding of the problem is such that it is tried to learn from data (rather than domain experts) which actions can be neglected for the design of the RL agent, i.e. a form of fixed action masking is learned. This can reduce the complexity of the agent and it can allow to remove undesired actions for probabilistic or online learning scenarios.

The contribution of the paper is the used of a monte carlo method to find the size of the action set, which is claimed as a Shapely-inspired algorithm, which is correctly spelled Shapley. Besides this mentioning of the Shapley inspiration there is no strong link or motivation for this connection.

Regarding the case study, I'm not sure we are actually solving a MDP here, the problem could similarly be addressed by a contextual bandit, if I understand it correctly. I'm also wondering what the incentive is to not always select the highest configuration, which should reduce the CPU utilization most and as costs seem not to be considered.

The paper is also missing a comparison with other techniques as baselines and a more thorough evaluation than the small case study experiment shown.

Finally, I found the writing style of the paper to be very hard to follow and confusing at times. The algorithms should be introduced more clearly in the text and there should be a focus on the novel contribution and its intuition, as well as some formal justification and analysis of the method.

**Summary Of The Paper:**

The paper addresses the problem of action set selection, i.e. identifying which actions should be available to a RL agent, during training. The set of available actions can influence the RL agent's performance or even hinder it to reach its goal.
A method to evaluate action sets is introduced and a case study is performed on a resource tuning example on cloud infrastructure.

**Summary Of The Review:**

The paper is confusing at some points and the actual contribution is only small.
There is no theoretical justification of the method.
The experimental evaluation has some shortcomings.
The paper is not ready to be published and needs a thorough revision.

---

> ### Author Response · Authors · 2021-11-23
> **Fixed the confusing parts**
>
> Thanks for your feedback. I have addressed your comments:
>
> 1. If one use the highest configuration all the time, the operating cost would be really high. That is why we are using RL to determine the optimal resource set point.
>
> 2. I have throughly revised the algorithm section to make the paper easier to understand.

---

> > ### Comment · Reviewer_TWbe · 2021-11-25
> > **Response**
> >
> > Thank you for revising the paper.
> > I acknowledge that it improves over the initial submission, but many of the concerns and limitations remain and I will maintain my score.

---

### Official Review · Reviewer_LXKk · 2021-11-02

**Correctness:** 3
**Technical Novelty And Significance:** 3
**Empirical Novelty And Significance:** 3
**Recommendation:** 3
**Confidence:** 3

**Details Of Ethics Concerns:**

I do not find any ethical concern with the work.

**Main Review:**

The paper solves an interesting and challenging problem in RL space that can have significant impact in overcoming one of the major challenges in RL which is to identify the optimal action space. The set of proposed algorithms provides a principled method to efficiently filter out unnecessary actions and discover a near optimal action set. Having said that I think paper in its current format is very hard to follow. The presentation needs to be improved significantly. It is challenging to understand the core of the technical contributions without a proper explanation. Algorithm 1-3 are explained in few sentences only. The list of unexplained variables or statements is exhaustive. To give a few examples: how state update function F is defined in algorithm 1? How the error margin and computational complexity is computed from state update function in algorithm 2? What is $a_i, b_i, c_i$ (note that c is used twice in same equation with different context) in algorithm 3? The empirical evaluation is performed in a real-world but new test environment. It does suggest a set of action recommendations with their cumulative reward value, but how to evaluate the performance of such actions in an RL setting is not clear to me. I also had a hard time in understanding Figure 3 and 4.

**Summary Of The Paper:**

The paper proposed a data-driven method for optimal action space selection in a reinforcement learning problem. Given a set of training state, action pair, the proposed approach first filters out the set of indispensable action set and then rank the other action set according to their cumulative reward values. To further improve the efficiency, a Monte Carlo sampling method is proposed for cut-off cardinality computation for action space. An action update rule is devised by computing optimal step size. Finally, a case study on a cloud environment is illustrated to select the optimal set of resources (#vCPUs and Memory size) to optimize the CPU utilization rate. The case study demonstrates that the Monte Carlo sampling based algorithm reduces action search space by 81% and then it creates a list of ranked action set. It is shown that a large action space does not necessarily lead to better performance for an RL agent.

**Summary Of The Review:**

The key idea of the paper is very interesting. Access to a data-driven methodology for selecting near-optimal action set in a generic and principled way would be very helpful in developing an RL based solution for a real-world challenging problem. However, I am afraid that the presentation of the paper is not up to mark at this moment; I failed to understand the key technical contributions and empirical evaluation methodology. Several important technical details and experimental settings are also missing.

---

> ### Author Response · Authors · 2021-11-23
> **Updated Presentation**
>
>
> Appreciate your feedback. I have updated the paper significantly:
>
> 1. The algorithm section has undergone a thorough revision.
> 2. A related work section has been added.

---

> > ### Comment · Reviewer_LXKk · 2021-11-29
> > **Thanks but the updated paper is still not up to the standard of ICLR**
> >
> > Thank you for providing an updated version of the paper; the presentation of the paper has indeed improved. Having said that, I completely agree with other reviewers that the quality of the paper in terms of its technical contributions and empirical evaluation still does not match the standard of ICLR. Therefore, I would maintain my scores.

---

### Decision · Program_Chairs · 2022-01-20

**Decision:**

Reject

**Comment:**

This paper proposes a method for finding the action space in reinforcement learning problems, characterizing the search space into dispensable and indispensable actions through a Monte Carlo approximation.

Reviewers are unanimous that the paper is not fit for publication at this stage. While it tackles an interesting problem and seems to be novel, the presentation leaves much to be desired; this area chair also had a hard time figuring out how the different parts of the paper fit together. Additionally, the use of a unique problem makes it hard to judge the contribution of the algorithm.